# Effects of Maillard Reaction Products on Skeletal Muscle Cells: An In Vitro Study Using C2C12 Myotubes

**DOI:** 10.3390/metabo15050316

**Published:** 2025-05-08

**Authors:** Marina Miyaki, Yusuke Komiya, Itsuki Sumiya, Rina Yamaguchi, Moeka Kuno, Chika Kojima, Ryosuke Makino, Takahiro Suzuki, Yoshihiro Suzuki, Issei Yokoyama, Keizo Arihara

**Affiliations:** 1Laboratory of Food Function and Safety, Department of Animal Science, School of Veterinary Medicine, Kitasato University, Towada 034-0021, Japan; miyaki.marina@st.kitasato-u.ac.jp (M.M.); itsukisumi75@gmail.com (I.S.); ag107093@gmail.com (R.Y.); moelovehorse.cafe@gmail.com (M.K.); qianjiaxiaodao515@gmail.com (C.K.); isseiyokoyama@agr.kyushu-u.ac.jp (I.Y.); arihara@vmas.kitasato-u.ac.jp (K.A.); 2Laboratory of Animal Nutrition, Department of Animal Science, Faculty of Agriculture, Iwate University, Morioka 020-8550, Japan; rmakino@iwate-u.ac.jp; 3Laboratory of Muscle and Meat Science, Department of Animal and Marine Bioresource Sciences, Research Faculty of Agriculture, Graduate School of Agriculture, Kyushu University, Fukuoka 819-0395, Japan; tsuzuki@agr.kyushu-u.ac.jp; 4Laboratory of Animal Health Science, Department of Animal Science, School of Veterinary Medicine, Kitasato University, Towada 034-0021, Japan; suzuyosh@vmas.kitasato-u.ac.jp

**Keywords:** Maillard reaction, MRPs, myotube, C2C12, muscle hypertrophy

## Abstract

**Background**: Maillard reaction products (MRPs) are known for their antioxidant properties; however, their effects on muscle cells remain unclear. This study aims to elucidate the effects of MRPs on muscle hypertrophy and atrophy in C2C12 myotubes. **Methods**: MRPs were prepared by heating L-lysine and D-glucose, and their antioxidant activity was assessed using the 2,2-diphenyl-1-picrylhydrazyl (DPPH) radical scavenging assay. Subsequently, mouse C2C12 myoblasts were cultured with MRPs until myotubes formed, and their diameters were measured to assess hypertrophic and atrophic changes. Akt phosphorylation was evaluated by Western blotting, and gene expression levels were analyzed via quantitative PCR. **Results**: The prepared MRPs exhibited high antioxidant activity in the DPPH radical scavenging assay. MRP treatment significantly increased the average myotube diameter by approximately 40% and enlarged the largest myotube diameter by up to 80%, potentially mediated by enhanced Akt phosphorylation. Under dexamethasone-induced atrophy, MRPs modestly attenuated the reduction in myotube diameter by approximately 20%, although the effect was not statistically significant, and did not significantly alter the fusion index either. Quantitative PCR analysis revealed that MRP treatment significantly reduced the mRNA expression of *Nfe2l2*, a key regulator of antioxidant response, whereas it had no notable effects on the expression of genes related to myoblast proliferation (*Myod1*), differentiation (*Myog*), hypertrophy (*Igf1*), atrophy (*Foxo1* and *Trim63*), and oxidative stress (*Cat*, *Gclc*, and *Nqo1*). **Conclusions**: Our findings suggested that MRPs possess antioxidant activity and promote myotube hypertrophy via Akt signaling. This study highlighted the potential of MRPs as functional ingredients for promoting muscle health, though further in vivo studies are required to validate their physiological relevance.

## 1. Introduction

Skeletal muscle is an essential tissue for metabolic and physical activity in animals. It exhibits a high degree of plasticity, allowing it to remodel its structure and function in response to various physiological and environmental stimuli, such as mechanical load, nutrient availability, and hormonal signals [1,2]. Reductions in mechanical load are well known to cause significant declines in muscle fiber size, leading to muscle atrophy. Situations such as prolonged bed rest, limb immobilization following fracture treatment, exposure to microgravity, or hind limb suspension in murine models are commonly used to study disuse-induced muscle atrophy [3,4]. Additionally, muscle wasting is frequently associated with aging and various neurological disorders [5,6]. In contrast, muscle hypertrophy occurs when skeletal muscle is exposed to increased mechanical load or stimulation, such as through resistance training or mechanical overload. This process involves an increase in the muscle fiber cross-sectional area, which enhances overall muscle strength and function.

The regulation of muscle mass involves a delicate balance between protein synthesis and degradation. Key molecular pathways, such as the Akt/mTOR pathway, promote muscle hypertrophy by stimulating protein synthesis [7], whereas the FoxO/atrogin-1 and MuRF1 pathways are associated with protein degradation and contribute to muscle atrophy [8,9]. Dysregulation of these mechanisms is a hallmark of muscle-wasting conditions like sarcopenia, cachexia, and glucocorticoid-induced atrophy, highlighting the need for interventions to maintain or improve muscle health.

In recent years, the role of food-derived bioactive compounds in modulating muscle mass has been reported. For example, branched-chain amino acids [10,11], omega-3 fatty acids [12], and plant-derived polyphenols [13] have demonstrated the ability to influence muscle protein metabolism and promote hypertrophy. These compounds often act by targeting oxidative stress, inflammation, or key molecular pathways regulating muscle size. Despite these advances, the exploration of food components as functional ingredients for muscle health is still in its infancy, and many potential candidates remain underexplored.

The Maillard reaction is a non-enzymatic reaction that occurs between amines (e.g., amino acids) and carbonyl groups (present in reducing sugars). This reaction often occurs during the processing or cooking of foods and produces a variety of chemicals. These chemicals, called Maillard reaction products (MRPs), contribute to the flavor, aroma, and color of foods [14,15]. Importantly, MRPs are known for their potent antioxidant properties [16], which can mitigate oxidative stress—a major factor in muscle atrophy and dysfunction. Despite extensive research on their chemical properties and health benefits, the biological effects of MRPs on skeletal muscle remain largely unknown. Given their prevalence in the human diet and their potential to influence cellular signaling, MRPs represent a promising but understudied candidate for promoting muscle health.

In light of these considerations, the aim of this study is to investigate the effects of MRPs on muscle hypertrophy and atrophy using a well-established in vitro model of C2C12 myotubes. To address this, we examine the antioxidant effects of the MRPs on muscle cells and determine whether MRPs promote myotube growth under normal conditions and mitigate muscle fiber shrinkage under dexamethasone-induced atrophic conditions. Furthermore, we investigate molecular-level changes by analyzing the expression of key factors related to muscle hypertrophy, atrophy, and oxidative stress in response to MRP treatment. Through this research, we seek to provide new insights into the functional role of MRPs in muscle biology and their feasibility as dietary interventions for muscle health.

## 2. Materials and Methods

To facilitate understanding of the overall experimental framework, we summarized the key dependent variables and their corresponding measurement methods in Table 1. Detailed descriptions of each method are provided in the following subsections. This study did not involve any experiments on human participants or animals and was, therefore, exempt from ethical approval requirements. All experiments were conducted at Kitasato University (Towada, Japan) between April 2021 and February 2023.

### 2.1. Preparation of Maillard Reaction Products (MRPs)

The MRPs were prepared as described previously [16]. Equimolar solutions (0.1 M) of L-lysine and D-glucose in 0.25% (*w*/*v*) sodium carbonate buffer (pH 11.0) were mixed and heated at 90 °C for 0, 15, 30, 45, 60, and 120 min in a dry thermos unit (DTU-1CN, TAITEC Co., Koshigaya City, Japan). Maillard reactions are known to proceed more efficiently under alkaline conditions [17] and, therefore, we performed the reaction under alkaline pH conditions, as described above, to obtain MRPs more efficiently. The intensity of the brown color increased with the progression of the reaction time (Figure 1A). After heating, the MRPs were immediately cooled on ice.

### 2.2. Evaluation of the Antioxidant Activity of MRPs

The antioxidant activity of MRPs was analyzed using the 2,2-diphenyl-1-picrylhydrazylradical (DPPH) antioxidant assay kit (Dojindo Laboratories, Kumamoto, Japan) according to the manufacturer’s instructions. Briefly, 20 μL of samples were added to each well of a microtiter plate, followed by 80 μL of assay buffer, 100 μL of ethanol, and 100 μL of DPPH working solution. A preliminary experiment was performed using MRP samples serially diluted ten-fold from the stock solution (i.e., 10^−1^ to 10^−5^). Based on the results, the undiluted stock solution was used for the final assay presented in this study. Then, the plate was incubated at 25 °C for 30 min in the dark. The optical density was measured at 520 nm using a Sunrise microplate absorbance reader. Vehicle (distilled water) and Trolox solution (80 μg/mL) were used as the control and positive control, respectively. The assay was conducted in triplicate.

### 2.3. Cell Culture

C2C12 cells were purchased from ATCC (CRL-1772, Manassas, VA, USA) and cultured, as described previously [18]. Briefly, a suspension of C2C12 myoblasts (30–32 passages) was seeded into 12-well plates coated with a collagen (Cellmatrix Type I-P; Nitta Gelatin, Osaka, Japan) at a cell density of 5.0 × 10^3^ cells per well and maintained in growth medium (Dulbecco’s modified Eagle’s medium (DMEM; Invitrogen, Grand Island, NY, USA) supplemented with 10% fetal bovine serum (FBS; Invitrogen), 1% antibiotic–antimycotic mixed stock solution (Nacalai Tesque Inc., Kyoto, Japan), and 0.5% gentamicin (Invitrogen)) until reaching confluence. After reaching confluence, the cells were cultured in differentiation medium (DMEM supplemented with 2% horse serum (HS; Invitrogen), 1% antibiotic–antimycotic mixed stock solution, and 0.5% gentamicin) for 4 days to form myotubes. All cells were incubated in a humidified atmosphere containing 5% CO_2_ at 37 °C. MRPs were supplemented in the differentiation medium at a concentration of 1%, and the medium was changed daily. Then, the average diameters of 50 myotubes were measured using ImageJ software (version 1.54g, Rasband W; National Institutes of Health).

In the experiment to assess the effect of MRPs on muscle atrophy, myotubes were treated with 1 μM of dexamethasone (DEX; Funakoshi Co., Ltd., Tokyo, Japan) with or without 1% MRPs for 14 days during the differentiation period.

### 2.4. Measurement of Intracellular Reactive Oxygen Species (ROS) Levels

Intracellular ROS levels of C2C12 myotubes were measured using the 2′,7′-dichlorofluorescein diacetate (H_2_DCFDA; Invitrogen) reagent. Cells were incubated with 20 μM of H_2_DCFDA for 30 min at 37 °C. After incubation, the cells were fixed with 4% paraformaldehyde (PFA) for 15 min at 4 °C. The cells were observed under a fluorescence microscope (BZ-X 810; KEYENCE, Osaka, Japan) and the fluorescence intensity (Ex: 450–490 nm; Em: 500–550 nm) was analyzed using the ImageJ software.

### 2.5. Western Blotting

Western blotting was performed, as described previously [19]. Briefly, C2C12 cells were homogenized in buffer containing 10% SDS, 40 mM of DTT, 5 mM of EDTA, and 0.1 M Tris-HCl (pH 8.0) with a protease inhibitor cocktail (Nacalai Tesque Inc.). The lysates were heated in boiling water for 3 min. Protein concentration was quantified using the Pierce BCA Protein Assay Kit (Thermo Fisher Scientific, Waltham, MA, USA) and adjusted to 8 mg/mL. For electrophoresis, 10 μL of protein sample was loaded onto a 10% SDS-polyacrylamide gel and run under reducing conditions, followed by transfer to PVDF membranes (Bio-Rad, Hercules, CA, USA). Membranes were blocked with 5% skim milk in TTBS for 45 min and then incubated overnight at 4 °C with primary antibodies diluted in CanGetSignal solution 1 (Toyobo, Osaka, Japan). The following antibodies were used: mouse monoclonal anti-actin (Chemicon MAB1501 (clone C4), 1:10,000), rabbit polyclonal anti-Akt (Cell Signaling, 9272, 1:1000), and rabbit monoclonal anti-phospho-Akt (pSer^473^, Cell Signaling, 4060 (clone D9E), 1:2000). Subsequently, membranes were incubated with HRP-conjugated secondary antibodies (anti-mouse IgG: Jackson ImmunoResearch, 1:5000; anti-rabbit IgG: Dako, 1:5000) diluted in CanGetSignal solution 2 (Toyobo). Protein bands were visualized using enhanced chemiluminescence reagents (ECL; GE Healthcare, Chicago, IL, USA), and band intensities were analyzed using ImageJ software.

### 2.6. RNA Isolation and qPCR Analysis

Quantitative real-time PCR analysis was performed, as described previously [19]. Briefly, total RNA was isolated from C2C12 cells using the ISOGENII reagent (NIPPON GENE, Tokyo, Japan), and complementary DNA (cDNA) was synthesized using SuperScript III Reverse Transcriptase (Invitrogen) and Oligo d (T)16 primers (Applied Biosystems, Waltham, MA, USA), following the manufacturers’ protocols. qPCR reactions were conducted on the Applied Biosystems StepOnePlus system using PowerUp SYBR Green Master Mix (Thermo Fisher Scientific). Primer specificity was confirmed through melting curve analysis. All primer sets were designed with ProbeFinder software (version 2.53, Roche Diagnostics) with an intron-spanning assay. The primer sets used in this study are listed in Table 2. Genes were analyzed using a standard curve constructed from a 4-point, 5-fold serial dilution (1-, 5-, 25-, and 125-fold) of complementary DNA aliquots pooled from one randomly chosen sample. All primer sets exhibited amplification efficiencies within the acceptable range (95–105%). TATA-box-binding protein (*Tbp*) was used as an internal standard.

### 2.7. Immunocytochemistry

Differentiated C2C12 myotubes were fixed in 4% paraformaldehyde (PFA) in PBS for 15 min at 4 °C. After fixation, the cells were permeabilized with 0.2% Triton X-100 for 15 min and subsequently blocked using 3% bovine serum albumin (BSA) in TPBS for 1 h at room temperature. The samples were then incubated overnight at 4 °C with a pan-specific monoclonal antibody against total myosin heavy chain (MyHC; clone MF20, MAB4470, R&D Systems) diluted 1:50, which recognizes all MyHC isoforms. After washing, the cells were treated for 1 h at room temperature with a rhodamine-conjugated anti-mouse IgG secondary antibody (KPL/SeraCare, Gaithersburg, MD, USA) diluted 1:250. Finally, the samples were mounted using ProLong™ Diamond Antifade Mountant containing DAPI (Thermo Fisher Scientific) and visualized using a BZ-X 810 fluorescence microscope (KEYENCE). We measured the diameters of 50 myotubes and selected the top 25 with the largest diameters according to previous studies [20,21], and the average diameter of them was considered as the representative diameter value of each well. These were designated as ‘largest myotubes’ in this study. The myotube fusion index was calculated as the percentage of myonuclei located in MyHC-positive myotubes containing two or more nuclei, relative to the total number of nuclei stained with DAPI, based on five randomly captured images per well. These analyses were performed using ImageJ software.

### 2.8. Statistical Analysis

Data are expressed as the mean ± standard error of the mean (SEM). Dunnett’s test was used to compare MRPs with vehicle values. For the dexamethasone treatment experiment, one-way ANOVA followed by the Tukey–Kramer multiple comparison test was conducted. Statistical significance was set at *p* < 0.05. All statistical analyses were performed using Excel-Toukei ver. 7.0 (Social Survey Research Information Co., Ltd., Tokyo, Japan). Graphs were created with Microsoft Excel (Redmond, WA, USA) and GraphPad Prism (version 10.4.1, GraphPad Software Inc., San Diego, CA, USA).

## 3. Results

### 3.1. Determination of Antioxidant Activity of MRPs

To determine the antioxidant activity of MRPs, a DPPH radical scavenging assay was performed. The MRPs exhibited approximately 55% DPPH radical scavenging activity at 60 min, which was nearly 3.0-fold higher than the non-heated control (Figure 1B). The scavenging activity progressively enhanced with the reaction time, peaking at 60 min, followed by a slight decline at 120 min. Notably, most MRPs exhibited a high scavenging activity comparable to that of the positive control, Trolox.

Subsequently, to elucidate whether MRPs influenced intracellular ROS levels, we examined the effect of MRPs on ROS levels in C2C12 myotubes by staining intramyocellular superoxide using H_2_DCFDA. The H_2_DCFDA-positive myotubes displayed green fluorescence, as shown in Figure 1C. MRPs significantly reduced intracellular ROS levels at 45, 60, and 120 min, with the 60 min treatment decreasing fluorescence intensity to approximately 57% of the vehicle group (Figure 1D). These findings indicated that MRPs possessed antioxidative activity and effectively reduced ROS levels in myotubes.

### 3.2. Effect of MRPs on C2C12 Myotube Diameter

To elucidate whether MRPs influenced myotube diameter, C2C12 myoblasts were cultured with MRPs for four days (Figure 2A). The myotube diameter was significantly increased in the MRP-treated groups (15, 45, 60, and 120 min) compared to the vehicle group, showing 1.36- to 1.53-fold increases depending on the treatment duration (Figure 2B). These findings suggested that MRPs promoted hypertrophy in cultured myocytes in vitro.

### 3.3. Effect of MRPs on Protein Synthesis Signaling Pathway and Muscle Fiber Types in C2C12 Myotubes

As MRPs increased the myotube diameter, we investigated the signal protein levels using Western blotting analysis to elucidate the signaling mechanisms underlying MRP-induced hypertrophy in C2C12 myotubes. We analyzed the phosphorylation levels of Akt protein, a key component of the major signaling pathway involved in protein synthesis [7]. Phosphorylation of Akt was elevated approximately 1.46-fold in the 60 min MRP group relative to the vehicle (Figure 3A,B). These findings suggested that MRPs promoted activation of protein synthesis pathways in myotubes.

Next, to determine the effect of MRPs on muscle fiber types, we analyzed the mRNA expression levels of *Myh* genes, markers of muscle fiber types, using real-time quantitative PCR. Slight changes in the expression of some *Myh* genes (*Myh2* and *Myh1*) were observed; however, no significant changes were observed in most groups (Figure 3C–F). This suggested that MRPs had little effect on muscle fiber types in myotubes.

### 3.4. Effect of MRPs on Dexamethasone-Induced Muscle Atrophy in C2C12 Myotubes

Because the 60 min MRPs induced significant muscle hypertrophy and Akt phosphorylation, they were selected for subsequent experiments. To investigate the potential of MRPs in counteracting muscle atrophy, C2C12 myoblasts were treated with dexamethasone (DEX), a well-established inducer of muscle atrophy [22]. Myotubes were immunofluorescent stained with an anti-total-MyHC antibody to analyze the myotube diameter and fusion index (Figure 4A). Since our previous study demonstrated that supplementation with MRPs increased the average myotube diameter (Figure 2B), we performed a more detailed analysis by generating a histogram of myotube diameters (Figure 4B). Notably, MRP treatment decreased the proportion of myotubes with diameters between 10 and 20 μm and increased the number of myotubes exceeding 80 μm compared to other groups. To further evaluate the effect on larger myotubes, which are more closely associated with muscle hypertrophy, we calculated the mean diameter of the top 25 largest myotubes (out of 50 measured). As a result, we found that MRP treatment increased the largest myotube diameter by approximately 1.78-fold compared to the vehicle group (Figure 4C). In contrast, the DEX treatment reduced the diameter to approximately 70% of the vehicle group. Co-treatment with MRPs partially restored the diameter, showing a 16% increase relative to the DEX group; however, this effect was not statistically significant. These results suggested that MRPs slightly suppressed dexamethasone-induced muscle atrophy, although their effect was not sufficient to completely recover the atrophy.

Next, to evaluate the effect of MRPs on myotube differentiation, we analyzed the fusion index. The MRP treatment did not enhance the fusion index compared to the vehicle group (Figure 4D). Similarly, there was no significant difference between the DEX, DEX+MRPs, and vehicle group. These findings suggested that MRPs did not influence myotube differentiation.

### 3.5. Effect of MRPs on mRNA Expression in C2C12 Myotubes

To elucidate the mechanism underlying the MRP-induced myotube hypertrophy and slight improvement of DEX-induced atrophy, mRNA expression levels of genes related to myoblast proliferation, differentiation, hypertrophy, atrophy, and oxidative stress were analyzed. No significant differences in *Myod1* and *Myog* expression were observed between the vehicle and MRPs groups. DEX treatment markedly suppressed the expression levels of *Myod1* (to 0.18-fold) and *Myog* (to 0.12-fold) compared to the vehicle group; however, MRP supplementation did not restore these reduced expression levels. The mRNA expression levels of growth factor *Igf1*, which is involved in muscle protein synthesis, and the atrophy-related genes *Foxo1* and *Trim63* are shown in Figure 5B. No significant differences in *Igf1* and *Foxo1* expression were observed among all groups. For *Trim63* expression, no significant differences were observed between the vehicle and MRPs groups; however, a significant increase was observed in the DEX and DEX+MRPs groups compared to the vehicle group (to approximately 3.7-fold). The mRNA expression levels of antioxidant-related genes—*Nfe2l2*, *Cat*, *Gclc*, and *Nqo1*—are shown in Figure 5C. A significant decrease in *Nfe2l2* expression was observed only in the MRPs group compared to the other groups (to 0.78-fold). For *Cat* and *Nqo1* expression, no significant differences were observed between the vehicle and MRPs groups; however, significant increases were observed in the DEX and DEX+MRPs groups compared to both the vehicle and MRPs groups. *Gclc* expression was not significantly different among all groups. These findings suggested that MRP treatment had little effect on mRNA expressions.

## 4. Discussion

We found that Maillard reaction product (MRP) treatment promoted muscle hypertrophy in C2C12 myotubes. Furthermore, this process was suggested to be induced through the phosphorylation of Akt protein, potentially due to the antioxidant effects of MRPs. However, MRPs were not potent enough to suppress dexamethasone-induced muscle atrophy. These findings suggested that while MRPs hold potential for promoting muscle hypertrophy under normal conditions, further investigation is needed to determine their effectiveness in mitigating muscle atrophy in pathological conditions. Additionally, the precise molecular mechanisms underlying these effects warrant further exploration to fully understand the potential of MRPs as functional food ingredients.

We have previously identified several functional properties of MRPs, including their antioxidant activity [15,23]. In this study, we investigated whether the antioxidant activity of MRPs affects muscle cell growth. Because oxidative stress negatively regulates muscle cell development [24,25], we hypothesized that MRPs would promote muscle growth. As expected, MRPs exhibited strong antioxidant activity and enhanced myotube hypertrophy. Reactive oxygen species (ROS), including superoxide radicals, hydrogen peroxide, hydroxyl radicals, and singlet oxygen, exist in various forms and function as key factors in biological defense mechanisms [26]. However, due to their high reactivity, ROS exhibit strong cytotoxicity. To counteract this, biological systems have developed mechanisms to rapidly scavenge ROS through antioxidant enzymes, such as superoxide dismutase [27,28]. When oxidative stress is exacerbated by factors such as lifestyle disturbances or disease, ROS production exceeds the capacity of antioxidant defenses, leading to excessive accumulation in the body. As a results, ROS attack cellular components, causing genetic damage, lipid peroxidation, and cell death, which in turn contribute to the development of various diseases, including cancer and inflammation [29,30].

In addition, ROS are involved in important biological reactions, such as signal transduction, regulation of gene expression, and prostaglandin synthesis [31,32]. A recent study exposed C2C12 myotubes to various reactive oxygen species (ROS) challenges, such as H_2_O_2_, diamide, and glucose oxidase, and found that the components of the Akt signaling pathway differed in their sensitivity to oxidative stress [33]. They showed that Akt was partially oxidized in untreated normal myotubes, and treatment with antioxidative catalase reduced Akt oxidation, leading to decreased interaction (and dephosphorylation) with PP2A, and hence increased Akt phosphorylation. Our findings were consistent with these observations, suggesting that basal oxidative stress present under normal conditions may mildly oxidize Akt, and that the antioxidant activity of MRPs reduces Akt oxidation, thereby enhancing its phosphorylation. In addition, ROS can target other key signaling pathways, such as mammalian target of rapamycin (mTOR) C1 [24] and ataxia–telangiectasia mutated protein kinase (ATM) [34], both of which are critical regulators of protein synthesis. The protein synthesis pathway is initiated when signaling factors, such as insulin and Igf-1, in the bloodstream bind to the insulin receptor substrate-1 (IRS-1) on the cell membrane, sequentially activating phosphoinositide 3-kinase (PI3K) and Akt. Activated Akt transduces signals to the mechanistic target of rapamycin complex 1 (mTORC1), which promotes protein synthesis by activating p70S6 kinase (p70S6K) and initiating mRNA translation. In addition, ATM directly phosphorylates hypoxia-inducible factor (HIF)1a, which also promotes tuberous sclerosis complex (TSC) 2 activity and, therefore, blocks mTORC1. It remains unclear whether these factors are affected by oxidative stress under basal conditions; however, if they are indeed oxidatively modified like Akt, the antioxidant activity of MRPs may directly improve their functional status. Even in the absence of overt oxidative stress, the increased phosphorylation of Akt could enhance the phosphorylation levels of downstream proteins, such as mTORC1 and p70S6K, ultimately contributing to the enhancement of protein synthesis and the observed increase in myotube diameter. In addition, MRP treatment significantly reduced the mRNA expression of *Nfe2l2*, which encodes the transcription factor Nrf2, a central regulator of the cellular antioxidant response. Nrf2 is activated under oxidative stress and enhances the expression of antioxidant enzymes while suppressing inflammatory mediators, thereby protecting cells from various stressors [35]. In skeletal muscle, Nrf2 deficiency has been reported to impair muscle regeneration following cardiotoxin-induced injury [36] and promote fiber-type transition from slow- to fast-twitch under spaceflight conditions [37]. The observed downregulation of *Nfe2l2* in our study may reflect a feedback mechanism by which the antioxidant properties of MRPs reduce oxidative stress levels, thereby diminishing the need for endogenous Nrf2 activation. However, how this reduction in Nrf2 expression relates to changes in muscle cell size remains unclear and warrants further investigation.

Furthermore, we investigated the inhibitory effects of MRPs on dexamethasone-induced muscle atrophy. The results indicated that MRPs slightly attenuated dexamethasone-induced muscle atrophy, but the effect was not sufficient. The two major proteolytic pathways involved in muscle atrophy are the ubiquitin-proteasome system and autophagy. In the ubiquitin-proteasome system, ubiquitin is conjugated to target proteins through a cascade of three catalytic enzymes: E1, E2, and E3. This ubiquitination serves as a degradation signal, directing the target proteins to the proteasome for degradation. Among these enzymes, E3 ubiquitin ligases play a crucial role by specifically recognizing substrates and facilitating their degradation. In skeletal muscle, MuRF1 and Atrogin-1 are well-known E3 ubiquitin ligases [38]. Their expression levels significantly increase during muscle atrophy, thereby promoting protein degradation. In this study, the mRNA expression of Trim63, the gene encoding MuRF1, was significantly upregulated by dexamethasone (DEX) treatment. However, no change in its expression was observed after administration of MRPs, suggesting that MRPs did not regulate muscle size via this pathway. One possible reason for the limited protective effect of MRPs against dexamethasone-induced atrophy is that glucocorticoid-mediated catabolism involves multiple pathways beyond oxidative stress, including direct activation of FoxO transcription factors and the glucocorticoid receptor [39]. Since the antioxidant activity of MRPs may only counteract ROS-related pathways, their efficacy might be enhanced under atrophic conditions, where oxidative stress plays a more central role. Future studies should explore MRP effects in other models of atrophy, such as aging- or denervation-induced muscle loss, where oxidative damage is a more prominent contributor.

We demonstrated that MRPs induce muscle hypertrophy, and several dietary compounds—such as polyphenols, including resveratrol [40,41], quercetin [42,43], and curcumin [44,45]—have been reported to exert protective effects on skeletal muscle by modulating oxidative stress and activating anabolic signaling pathways. While the mechanisms vary, these compounds commonly converge on the Akt/mTOR pathway or reduce catabolic signaling. Similarly, our findings suggested that MRPs may serve as a novel dietary compound with protective effects on skeletal muscle, comparable to polyphenols. In contrast, compounds related to MRPs, known as advanced glycation end products (AGEs), have been reported to inhibit muscle growth. Suzuki et al. reported that AGE treatment inhibited cell proliferation in C2C12 myoblasts [46]. Additionally, mice receiving a diet high in AGEs showed lower muscle weight and lower phosphorylation levels of p70S6K relating to protein synthesis [47]. Furthermore, AGEs promoted ROS production in skeletal muscle cells through the phosphorylation of PKCα and p47phox [48]. These reports described phenomena that were the opposite of our findings in this study on MRPs. Both AGEs and MRPs are formed through the Maillard reaction, a non-enzymatic glycation process that occurs between reducing sugars and proteins. While AGEs are a subset of MRPs, they are particularly known for accumulating in the body and promoting inflammation and aging. In the above studies, the tested compounds were classified as AGEs, a group of Maillard reaction products formed during the later stages of the reaction. AGEs can be generated endogenously through physiological processes, such as oxidative stress and aging, or exogenously through food processing following the progression of the Maillard reaction. They generated AGEs using a method where 50 mg/mL of bovine serum albumin (BSA) was incubated at 37 °C with 0.1 M glyceraldehyde in 0.2 M phosphate buffer (pH 7.4) for seven days [49]. In contrast, we prepared MRPs using the following method: equimolar solutions (0.1 M) of L-lysine and D-glucose in 0.25% (*w*/*v*) sodium carbonate buffer (pH 11.0) were mixed and heated at 90 °C for 0, 15, 30, 45, 60, and 120 min, suggesting that the resulting compounds may differ. Although the specific composition of the generated products remains unclear, elucidating these differences may provide insights into their distinct effects on muscle cells.

In this study, MRPs exhibited antioxidant activity and were suggested to promote myotube hypertrophy through Akt signaling. Additionally, MRPs were found to slightly attenuate dexamethasone-induced muscle atrophy. However, several limitations remain, particularly regarding the differences between MRPs and AGEs’ effects. Further research is required to elucidate how differences in preparation conditions influence the composition of the resulting compounds. Moreover, the bioactive components within the generated MRPs that contribute to muscle hypertrophy have yet to be identified, necessitating further investigation. Future studies will examine the effects of these individual compounds on skeletal muscle cells. Additionally, as this study was conducted using an in vitro muscle cell model, in vivo studies using mice will be necessary to further elucidate the physiological effects of MRPs in a living system.

## 5. Conclusions

MRPs effectively increased the muscle fiber diameter through Akt signaling in C2C12 myotubes, potentially due to the antioxidant properties. These findings suggested that MRPs may have potential as a functional ingredient for supporting muscle mass maintenance and promoting skeletal muscle hypertrophy. However, the limited ability of MRPs to counteract dexamethasone-induced atrophy may be due to the fact that dexamethasone induces atrophy through a broad range of mechanisms, whereas MRPs primarily exert their effects through antioxidant pathways. Therefore, using muscle atrophy models induced by oxidative stress could highlight the effects of MRPs more clearly, and further research is necessary. Additionally, the specific functional components of MRPs remain unidentified, which warrants further investigation to better understand their potential in muscle health applications.

## Figures and Tables

**Figure 1 metabolites-15-00316-f001:**
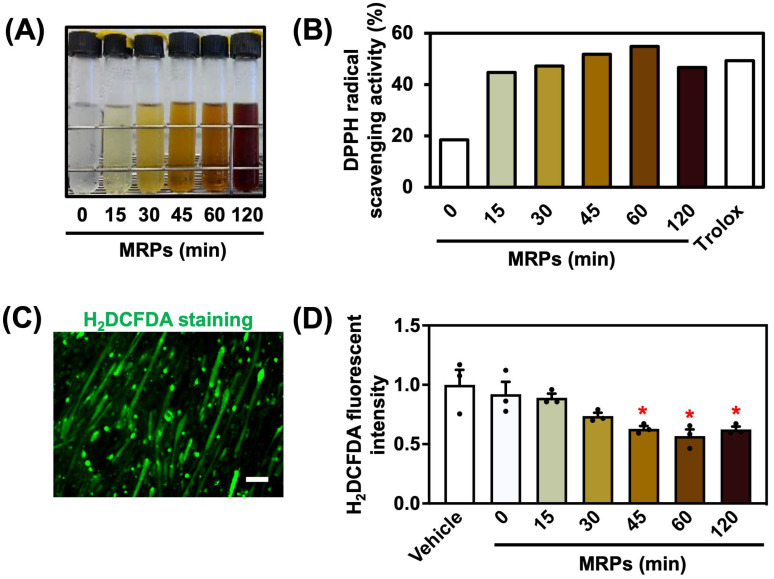
Antioxidant activity of Maillard reaction product (MRPs): (**A**) Visual appearance of Maillard reaction products (MRPs) at different heating times (0, 15, 30, 45, 60, and 120 min). (**B**) The 2,2-diphenyl-1-picrylhydrazylradical (DPPH) radical scavenging activity (%) of MRPs. Trolox (80 μg/mL) was used as a positive control. (**C**) Fluorescence image of the 2′,7′-dichlorofluorescein diacetate (H_2_DCFDA)-stained C2C12 myotubes. The scale bar indicates 50 µm. (**D**) Quantification of intracellular H_2_DCFDA fluorescence intensity. Data are means ± SEM (n = 3 independent cultures, * *p* < 0.05 compared with the vehicle controls).

**Figure 2 metabolites-15-00316-f002:**
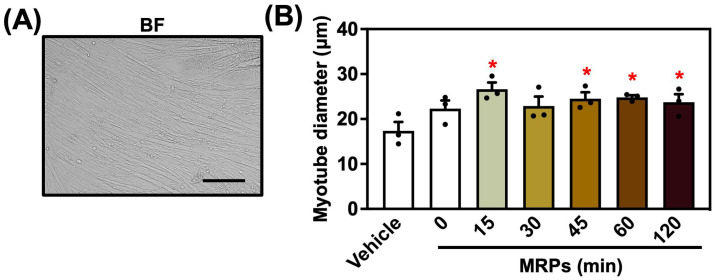
Effects of Maillard reaction product (MRPs) on C2C12 myotube diameter: (**A**) Bright-field (BF) image of differentiated C2C12 myotubes. The scale bar indicates 50 µm. (**B**) Quantification of myotube diameters. Data are means ± SEM (n = 3 independent cultures, * *p* < 0.05 compared with the vehicle controls).

**Figure 3 metabolites-15-00316-f003:**
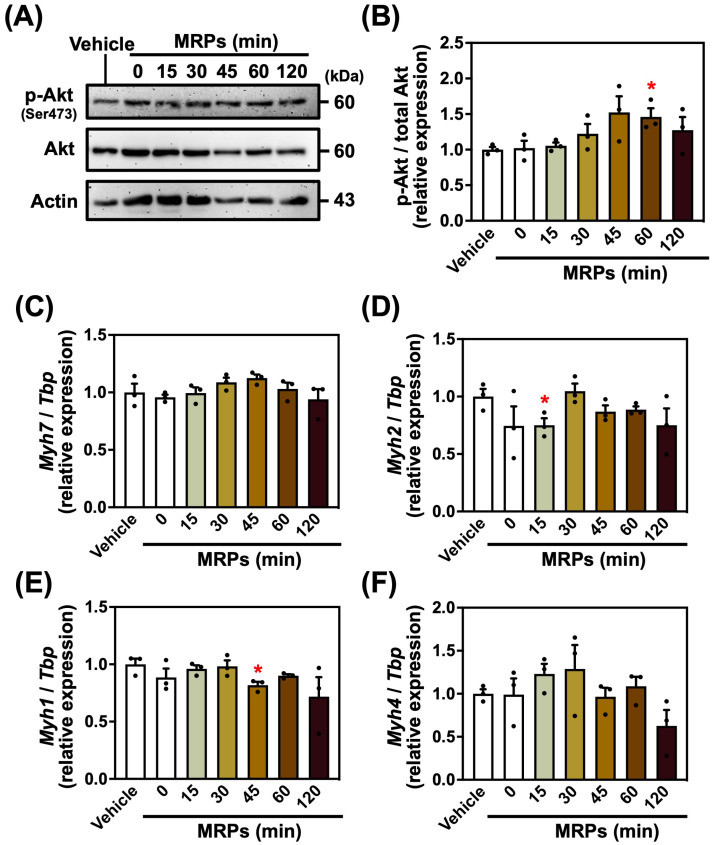
Effect of Maillard reaction products (MRPs) on protein synthesis signaling pathways and muscle fiber types in C2C12 myotubes: (**A**) Western blot analysis of phosphorylated Akt (p-Akt at Ser473), total Akt, and actin (loading control) in C2C12 myotubes treated with MRPs. (**B**) The densitometry quantification of images of panel (**A**). The p-Akt to total Akt ratio was quantified. (**C**–**F**) Relative mRNA expression levels of myosin heavy-chain isoforms (*Myh7*, *Myh2*, *Myh1*, and *Myh4*) normalized to TATA-box-binding protein (*Tbp*) in C2C12 myotubes treated with MRPs. Data are means ± SEM (n = 3 independent cultures, * *p* < 0.05 compared with the vehicle controls).

**Figure 4 metabolites-15-00316-f004:**
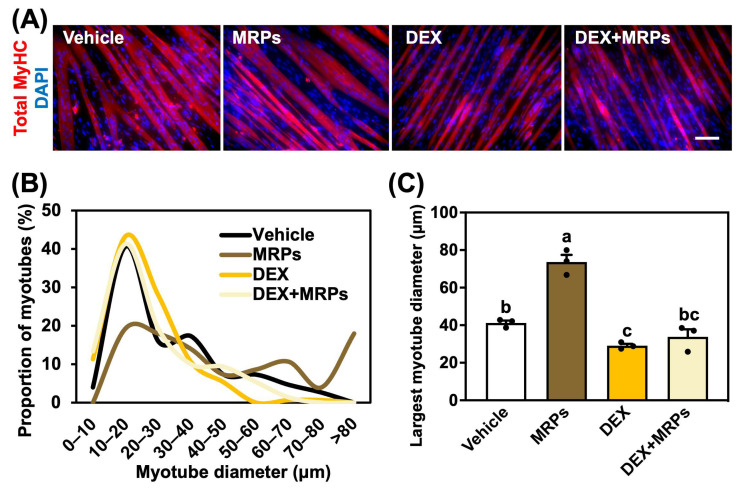
Effect of Maillard reaction products (MRPs) on the myotube diameter and fusion index in C2C12 myotubes treated with dexamethasone (DEX): (**A**) Immunofluorescence images of C2C12 myotubes stained with anti-total-MyHC (red) and DAPI (blue). The scale bar indicates 50 µm. (**B**) Distribution of myotube diameters. The diameters of 50 myotubes per group were measured, and the proportion of myotubes within each diameter range was plotted for each treatment condition. (**C**) Quantification of the largest 25 myotube diameters, out of 50 measured. (**D**) Myotube fusion index, defined as the ratio of myonuclei in myotubes containing two or more nuclei. Data are means ± SEM (n = 3 independent cultures, different superscripts indicate a significant difference among the four groups).

**Figure 5 metabolites-15-00316-f005:**
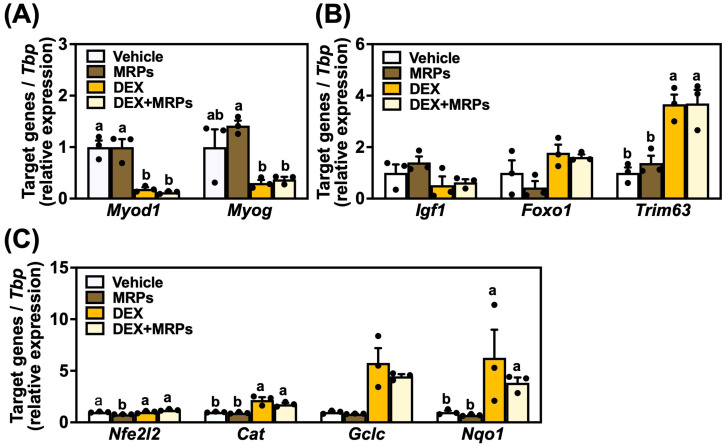
Effect of Maillard reaction products (MRPs) on mRNA expression of genes related to myoblast proliferation, differentiation, hypertrophy, atrophy, and oxidative stress: (**A**) relative mRNA expression levels of *Myod1* and *Myog*, (**B**) relative mRNA expression levels of *Igf1*, *Foxo1*, and *Trim63*, and (**C**) relative mRNA expression levels of *Nfe2l2*, *Cat*, *Gclc*, and *Nqo1*. All gene expression levels were normalized to TATA-box-binding protein (*Tbp*). Data are means ± SEM (n = 3 independent cultures, different superscripts indicate a significant difference among the four groups).

**Table 1 metabolites-15-00316-t001:** Summary of dependent variables and corresponding measurement instruments/methods.

Dependent Variable	Measurement Instruments/Methods
Maillard reaction products	Reaction of 0.1 M L-lysine and D-glucose in 0.25% (*w*/*v*) sodium carbonate buffer (pH 11.0)
Average myotube diameter	Microscopy and ImageJ analysis; measured the diameters of 50 individual myotubes
Largest myotube diameter	Microscopy and ImageJ analysis; top 25 largest myotubes selected from 50 measured
Fusion index	Immunostaining with anti-MyHC antibody; percentage of myonuclei in MyHC-positive myotubes (≥2 nuclei)
Akt phosphorylation	Western blotting
mRNA expression	Quantitative real-time PCR
Antioxidant activity	DPPH radical scavenging assay; H_2_DCFDA staining

MyHC, myosin heavy chain; DPPH, 2,2-diphenyl-1-picrylhydrazylradical; H_2_DCFDA, 2′,7′-dichlorofluorescein diacetate.

**Table 2 metabolites-15-00316-t002:** List of primer sequences for RT-qPCR.

Gene	Forward (5′-3′)	Reverse (5′-3′)
*Cat*	CCTTCAAGTTGGTTAATGCAGA	CAAGTTTTTGATGCCCTGGT
*Foxo1*	GTGGGGCAACCTGTCGTA	TTCTCGGCTGAGCTCTCG
*Gclc*	AGATGATAGAACACGGGAGGAG	TGATCCTAAAGCGATTGTTCTTC
*Igf1*	GCAGTTCTAACACCAGCCCA	CCCACTCGATCGTACCTTCTG
*Myh1*	TCGCTGG CTTTGAGATCTTT	CGAACATGTGGTGGTTGAAG
*Myh2*	AAAGCTCCAAGGACCCTCTT	AGCTCATGACTGCTGAACTCAC
*Myh4*	GTCACCAAAGGCCAGACG	ACATCTTCTCATACATGGACTTGG
*Myh7*	GAGCAGCAGGTGGATGATCT	GCTTGGCTCGCTCTAGGTC
*Myod1*	AGCACTACAGTGGCGACTCA	GGCCGCTGTAATCCATCAT
*Myog*	CCTTGCTCAGCTCCCTCA	TGGGAGTTGCATTCACTGG
*Nfe2l2*	CATGATGGACTTGGAGTTGC	CCTCCAAAGGATGTCAATCAA
*Nqo1*	AGCGTTCGGTATTACGATCC	AGTACAATCAGGGCTCTTCTCG
*Tbp*	GGGGAGCTGTGATGTGAAGT	CCAGGAAATAATTCTGGCTCAT
*Trim63*	TGACATCTACAAGCAGGAGTGC	TCGTCTTCGTGTTCCTTGC

*Cat*, catalase; *Foxo1*, forkhead box O1; *Gclc*, glutamate-cysteine ligase catalytic subunit; *Igf1*, insulin-like growth factor 1; *Myh1*, myosin heavy chain 1; *Myh2*, myosin heavy chain 2; *Myh4*, myosin heavy chain 4; *Myh7*, myosin heavy chain 7; *Myod1*, myogenic differentiation 1; *Myog*, myogenin; *Nfe2l2*, nuclear factor erythroid-derived 2-like 2; *Nqo1*, NAD(P)H dehydrogenase quinone 1; *Tbp*, TATA-box-binding protein; *Trim63*, tripartite-motif-containing 63.

## Data Availability

The original contributions presented in the study are included in the article. Further inquiries can be directed to the corresponding author.

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
