# Peer review of "Effects of Maillard Reaction Products on Skeletal Muscle Cells: An In Vitro Study Using C2C12 Myotubes"

_metabolites, 2025, doi:10.3390/metabo15050316_

Round 1
Reviewer 1 Report
Comments and Suggestions for Authors
Lysine-glucose Maillard reaction products promote skeletal muscle hypertrophy in C2C12 myotubes
General comments:
The manuscript addresses an interesting aspect of the role of MRPs in muscle hypertrophy. However, there are several areas that require improvement to enhance the overall quality of the manuscript. Provide a more in-depth interpretation of the results in the discussion section and eliminate vague terms. Additionally, revise the conclusion section to accurately reflect the study's findings, and clearly state the limitations of the study.
Abstract:
Line 27: Provide a little bit more implications of the minimal results.
Mention the percentage or values for, e.g., a 20% increase in diameter, etc., to enhance the reader's interest.
Modify the conclusion by suggesting further validation in vivo.
Introduction:
Line 35: the sentence “It is a dynamic tissue with unique characteristics” is vague. Rephrase it
The objective of the study hasn't been clearly addressed. Clearly and directly describe the objective of the study.
Materials and methods:
Line 80: Provide the reference for why sodium carbonate of 0.25 buffered at pH 11.
The control looks missing, was the blank prepared?
The dilution and concentration of MRPs have not been mentioned.
Moreover, mention how many replicates/samples were used
The catalog no for collagen and passage no. of C2C12 cells has not been mentioned
In Western bloat, mention the protein loading amount per lane.
In the section on RNA isolation, mention the details of the standard curve, including the no. of dilution and efficiency calculation.
Mention the criteria for choosing 25 large myobites
Results:
Significantly increased mentioned too often, mentioned the fold or percentage increase allover where applicable.
Discussion:
The discussion lacks a deep analytical interpretation of the findings.
Provide an in-depth discussion related to MRPs improving muscle atrophy.
Conclusion:
Revise the conclusion section to truly depict the study findings and mention broader implications and further recommendations.
Line 382: Enhance their therapeutic efficacy” is vague; rephrase it.
Reviewer 2 Report
Comments and Suggestions for Authors
I would like to congratulate the authors for their effort in identifying the effect of Maillard reaction products on muscle health.
I would kindly like to offer a few comments:
- Title: I wouldn't approach this as a conclusion, but rather a title like "effect of Maillard reaction products (lysine-glucose) on muscle cells," and I would also add the study design.
- Abstract: It would be clearer if they divided the sections into background, methods, results, and conclusions. They would also include the meaning of all the acronyms in parentheses.
- Introduction: I find it clear and it encourages reading the manuscript.
- Methods: I think it would be easier to understand the study if they added a table with the different dependent variables and their corresponding measurement instrument/method.
- It might also be useful to add why ethics committee approval is required, the context, date, and location where the research was conducted.
- Results: I think they are well expressed.
-Discussion: I miss a comparison of your findings with those of other researchers.
-Conclusion: I find it clear and consistent with the objectives.
Round 2
Reviewer 1 Report
Comments and Suggestions for Authors
The authors have satisfactorily addressed all the comments raised in the first round of revision. The study is now acceptable in its current form.